# Livestock abortion surveillance in Tanzania reveals disease priorities and importance of timely collection of vaginal swab samples for attribution

Felix Lankester[1,2]*, Tito J Kibona[2], Kathryn J Allan[3], William de Glanville[3], Joram J Buza[4], Frank Katzer[5], Jo E Halliday[3], Blandina T Mmbaga[6], Nick Wheelhouse[7], Elisabeth A Innes[5], Kate M Thomas[8], Obed M Nyasebwa[9], Emanuel Swai[9], John R Claxton[3], Sarah Cleaveland[3]

[1]Paul G. Allen School for Global Health, Washington State University, Pullman, United States; [2]Global Animal Health Tanzania, Arusha, United Republic of Tanzania; [3]School of Biodiversity, One Health, and Veterinary Medicine, College of Medical, Veterinary and Life Sciences, University of Glasgow, Glasgow, United Kingdom; [4]Nelson Mandela African Institution of Science and Technology, Arusha, United Republic of Tanzania; [5]Moredun Research Institute, Pentlands Science Park, Edinburgh, United Kingdom; [6]Kilimanjaro Clinical Research Institute, Moshi, United Republic of Tanzania; [7]School of Applied Sciences, Edinburgh Napier University, Edinburgh, United Kingdom; [8]Centre for International Health, University of Otago, Dunedin, New Zealand; [9]Ministry of Livestock and Fisheries, Dodoma, United Republic of Tanzania

*For correspondence:
felix.lankester@wsu.edu

Competing interest: The authors declare that no competing interests exist.

## eLife assessment

This **important** study reports the use of a surveillance approach in identifying emerging diseases, monitoring disease trends, and informing evidence-based interventions in the control and prevention of livestock abortions, as it relates to their public health implications. The data support the **convincing** finding that abortion incidence is higher during the dry season, and occurs more in cross-bred and exotic livestock breeds. Aetiological and epidemiological data can be generated through established protocols for sample collection and laboratory diagnosis. These findings are of potential interest to the fields of veterinary medicine, public health, and epidemiology.

**Abstract** Lack of data on the aetiology of livestock diseases constrains effective interventions to improve livelihoods, food security and public health. Livestock abortion is an important disease syndrome affecting productivity and public health. Several pathogens are associated with livestock abortions but across Africa surveillance data rarely include information from abortions, little is known about aetiology and impacts, and data are not available to inform interventions. This paper describes outcomes from a surveillance platform established in Tanzania spanning pastoral, agropastoral and smallholder systems to investigate causes and impacts of livestock abortion. Abortion events were reported by farmers to livestock field officers (LFO) and on to investigation teams. Events were included if the research team or LFO could attend within 72 hr. If so, samples and questionnaire data were collected to investigate (a) determinants of attribution; (b) patterns of events, including species and breed, previous abortion history, and seasonality; (c) determinants of reporting, investigation and attribution; (d) cases involving zoonotic pathogens. Between 2017–2019, 215 events in cattle (n=71), sheep (n=44), and goats (n=100) were investigated. Attribution,

achieved for 19.5% of cases, was significantly affected by delays in obtaining samples. Histopathology proved less useful than PCR due to rapid deterioration of samples. Vaginal swabs provided practical and sensitive material for pathogen detection. Livestock abortion surveillance, even at a small scale, can generate valuable information on causes of disease outbreaks, reproductive losses and can identify pathogens not easily captured through other forms of livestock disease surveillance. This study demonstrated the feasibility of establishing a surveillance system, achieved through engagement of community-based field officers, establishment of practical sample collection and application of molecular diagnostic platforms.

## Introduction

Livestock reproductive losses, including spontaneous abortion, are a major concern for the livestock industry worldwide, resulting in significant economic loss and posing a threat to public health (*Food and Agriculture Organization, 2011*). The impacts of livestock abortion on the world's poorest livestock-keepers, who are heavily dependent on livestock for food security and livelihoods (*Food and Agriculture Organization, 2011*), is likely to be substantial. For these families, the loss of a livestock foetus and a subsequent decline in milk production reduces the availability of a high-quality food source (milk) that can be essential for childhood growth and cognitive development (*Neumann et al., 2003*). In addition, livestock reproductive losses reduce income (from sales of meat and milk), and cause a loss of livestock assets that are a critical source of wealth, collateral or a safety net in times of need (*The World Bank, 2021*). Recent analyses have shown that because of the need for increased spending on animal management, livestock abortions mayalso have indirect negative impacts on household expenditure and education (*Haseeb et al., 2019*). Livestock abortion can also pose a direct threat to human health because many abortigenic agents are also zoonotic (*Givens and Marley, 2008*; *Thomas et al., 2022*).

Surveillance is defined as the real-time (or near real-time) collection, analysis, interpretation, and dissemination of health-related data to enable the early identification of the impact (or absence of impact) of potential health threats, which require effective action (*Hulth, 2014*). Effective livestock health surveillance provides critical data for evidence-based approaches to livestock disease control and management but this requires reliable, high-quality, and timely data that can be drawn from multiple sources (*George et al., 2021*). Over the past decade, increasing attention has been given to animal health syndromic surveillance (*Dórea and Vial, 2016*), which relies on detection of health indicators, such as livestock abortion, that are discernible before a confirmed diagnosis is made. However, systematic reviews of the literature that collectively cover a period from 2000 to 2016 indicate that syndromic surveillance programmes have mostly been implemented in Europe, North America or Australasia with only a single pilot project identified in Africa (*Dórea and Vial, 2016*; *Dórea et al., 2011*).

In public health, event-based surveillance has also been gaining attention for the early detection of unusual events that might signal acute and emerging human health risks (*World Health Organization, 2008*; *Africa Centres for Disease Control and Prevention, 2018*). This involves the collection, analysis and interpretation of information through both formal and informal channels to rapidly identify unusual or unexpected health events, such as disease outbreaks, emerging infectious diseases, or other public health threats. Although the WHO guidelines for event-based surveillance (*World Health Organization, 2008*) make explicit reference to the capture of information around unusual disease events in animals, the development of integrated early-warning systems involving animal disease events is still very limited. Livestock surveillance has a clear potential for identifying and preventing outbreaks of zoonotic and emerging diseases; a substantial proportion (39.4%) of livestock pathogens infect humans (*Cleaveland et al., 2001*), and ungulates harbour more zoonotic pathogens than any other taxonomic group (*Woolhouse and Gowtage-Sequeria, 2005*). With several abortigenic agents also causing emerging livestock diseases, for example those caused by Schmallenberg virus, bluetongue virus and porcine reproductive and respiratory syndrome virus, livestock abortion may represent a particularly important syndromic target for zoonotic and emerging disease surveillance. However, there is limited information available on current practices, effectiveness, and challenges/opportunities of livestock abortion surveillance, particularly in low- and middle-income countries.

**eLife digest** Livestock reproductive losses are a major concern for farmers worldwide as they cause significant economic impacts, particularly for those that are heavily dependent on their livestock for food security. On top of this, such losses can also pose a threat to public health if they are caused by infections that can also be transmitted to humans.

Spontaneous abortion (when a pregnancy ends early and a foetus is expelled) can be caused by a number of factors, including infections, nutritional deficiencies and genetic issues. Identifying the cause is easier if high quality samples are collected from the aborting mother and the foetus. However, this can be difficult in some low-and middle-income countries, where such samples are rarely collected and analysed.

Lankester et al. wanted to investigate whether livestock abortion surveillance could be used to understand the causes and effects of livestock abortion in Tanzania. To do this, the researchers asked farmers to report abortion cases to livestock field officers. These officers alerted investigation teams to collect samples and conduct questionnaires which provided information on the livestock breeds, seasonal patterns and potential pathogens involved in 215 abortion cases in cattle, sheep and goats.

Analysis revealed that successfully identifying the cause of abortion depends heavily on the timing and quality of the samples. The chances of diagnosis decreased with each day that passed between the abortion and the samples being collected. Vaginal swabs, which are easier to collect than those from the placenta or aborted foetus, were the most effective at detecting abortion-causing infectious agents.

The analysis also revealed that many of the livestock which had an abortion in the previous 12 months had experienced one or more abortions before. This suggests that an infectious agent may be the cause and that, through surveillance and accurate diagnosis, managing these animals by removing them from the herd might improve productivity. Abortions were also more common in non-local breeds of cattle and goats, suggesting that local breeds may have a degree of resistance to abortion.

The findings of Lankester et al. reveal a method of livestock surveillance that is feasible in areas with limited resources and could be used to increase understanding of the causes of livestock abortion. Such information could help to direct interventions that prevent abortion and improve livestock health, ultimately helping to improve food security while reducing the risk of infection for livestock-owners in lower- and middle-income countries.

---

Livestock abortion has many causes, including infectious, nutritional, physical, and genetic factors. The diversity of causative agents and variation in the relative importance of agents across different livestock management systems and geographies makes abortion diagnosis complex and challenging (*Wolf-Jäckel et al., 2020*). Challenges relate to timely collection of diagnostic samples, sample availability and deterioration, biases in detectability of agents, as well as complexities around establishment of a causal aetiology, particularly for pathogens such as *Coxiella burnetii* and *Escherichia coli*, which can often be present as incidental infections (*Thomas et al., 2022*; *Wolf-Jäckel et al., 2020*). As a result, aetiological attribution of livestock abortion rarely exceeds 35%, even in well-resourced industrialised livestock systems (*Cabell, 2007*; *Wolf-Jäckel et al., 2020*) and data are particularly sparse in African livestock systems.

A recent prospective study of the aetiologies of livestock abortion, carried out in northern Tanzania (*Thomas et al., 2022*), investigated 215 cases of livestock abortion of which an attribution was made in 41 cases (19%). The infectious causes of abortion were identified as Rift Valley fever virus (RVFV) in 14 cases (6.6%), followed by *Neospora caninum* in 10 (4.7%), pestiviruses in 9 (4.2%), *Coxiella burnetii* in 6 (2.8%), and *Brucella* sp., and *Toxoplasma gondii* in one case each (0.5%). Our study draws on the operational data generated from establishment of a livestock abortion study to examine characteristics of reporting and investigation of cases of livestock abortion. While this study was not designed as a systematic or comprehensive evaluation of a surveillance system, we present data on several key attributes (*ECDC, 2014*) of the platform, including simplicity, data quality, representativeness, timeliness, and usefulness. We discuss our findings in relation to the feasibility, practicality, and value of establishing a livestock abortion surveillance framework to support evidence-based interventions to improve livestock development, livelihoods, and human health in Africa.

**Table 1.** The number (and percentage) of abortion cases by species and agro-ecological zone and the composition of the livestock herds (and percentage) in investigated households.

| Event | Category | Number (%) |
|---|---|---|
| | All species | 215 |
| | Cattle | 71 (33%) |
| | Goats | 100 (46.5%) |
| Number of abortion cases | Sheep | 44 (20.5%) |
| | Pastoral | 144 (67.0%) |
| | Agro-pastoral | 1 (0.5%) |
| Number of abortion cases in each agricultural ecological zone | Peri-urban | 70 (32.5%) |
| Number of households that had an abortion event | Households | 150 |
| | Pastoral | 84 (56.0%) |
| | Agro-pastoral | 1 (0.7%) |
| Number of households in each agricultural ecological zone | Peri-urban | 65 (43.3%) |
| Composition of the 150 herds | Cattle, goats and sheep | 77 (51.3%) |
| | Cattle and goats | 17 (11.3%) |
| | Goats and sheep | 7 (4.7%) |
| | Cattle and sheep | 1 (0.7%) |
| | Cattle only | 40 (26.6%) |
| | Goats only | 8 (5.3%) |

## Results

### Descriptive statistics

Between October 2017 and September 2019, 215 abortion cases were reported from 150 households in 13 of the 15 target wards. The distribution of investigated cases in relation to agro-ecological system and herd/flock composition is shown in *Table 1*. Out of the 150 households investigated, most (n=115) had only one event investigated. Of the remaining households, 21 had two cases investigated, eight had three, two had four, three had five and one household had 11 cases investigated. Herd level summary statistics have been provided in *Supplementary file 1*.

The sensitivity of the platform (the percentage of expected abortion cases that were investigated) ranged from 0% to 12.4% for cattle, 0–1.2% for goats and 0–0.3% for sheep. A higher percentage of expected abortions in cattle were reported in smallholder wards (2.7%) than in other wards (0% for agropastoral and 0.31% for pastoral wards), with particularly high reporting in one smallholder ward, Machame Mashariki, close to Moshi town, where 12.4% of expected cattle abortions were investigated.

Over the 12 months preceding a reported abortion event, abortions, peri-natal mortalities, or stillbirths occurred in 52 of 184 (28.3%) cattle herds, 98 of 168 (58.3%) goat herds and 93 of 140 (66.4%) sheep herds that had at least one adult female. Out of herds that had at least one abortion case, the mean (median, range) number of cases of abortion per herd in the past 12 months was 3.7 (3, 1 – 12) for cattle, 9.0 (6, 1 – 70) for goats and 6.7 (4, 1 – 67) for sheep.

### Determinants of investigation

The number of cases reported to the research team by each LFO varied considerably (median = 5, range 0–84). Of the 215 cases, 70% were reported by three (20%) of the LFOs, with one reporting 84 cases (39.1%). Two LFOs did not report a single case (*Figure 1a*). The range in the interval between the report and the subsequent investigation by the research team was 0 to 4 days with a median of 1 day (*Figure 2*).

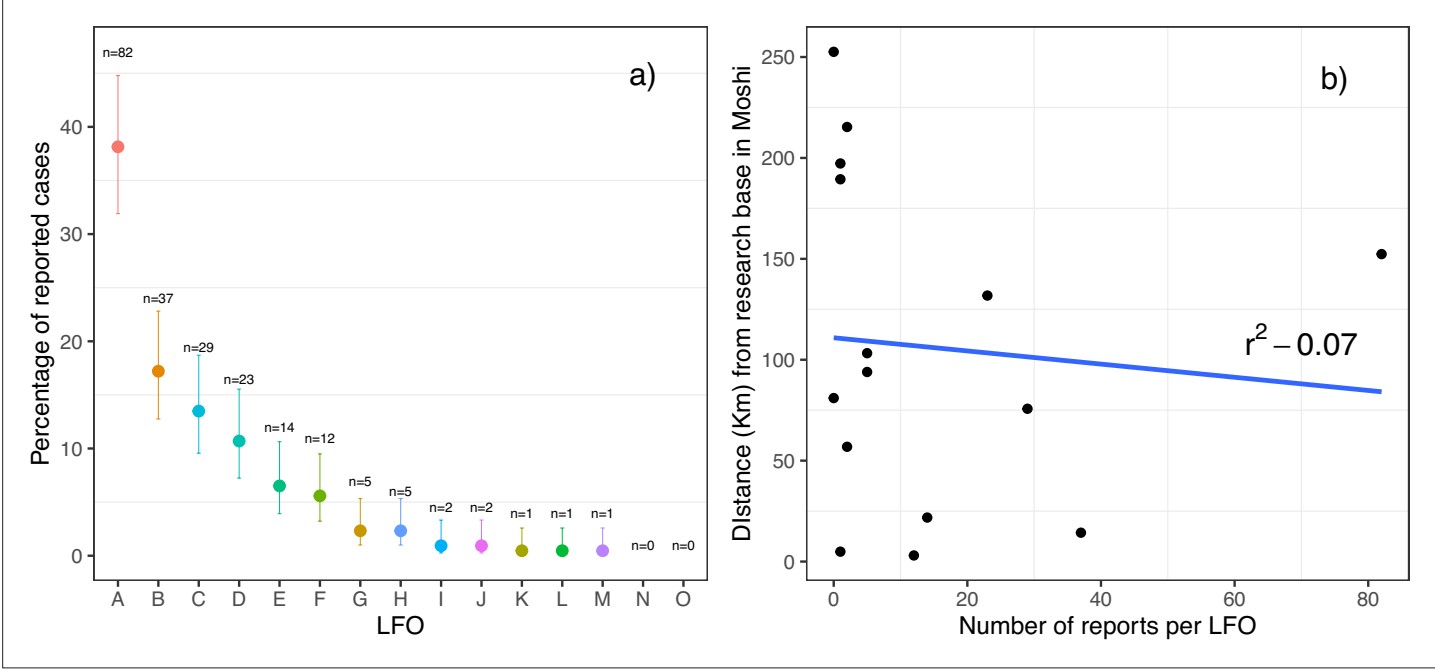

**Figure 1.** Llvestock field officer reports. (**a**) The number and percentage of abortion cases reported by each LFO (95% error bars, n = 215) (**b**) The relationship between the number of reports per LFO and the distance to the research laboratories based in the town of Moshi.

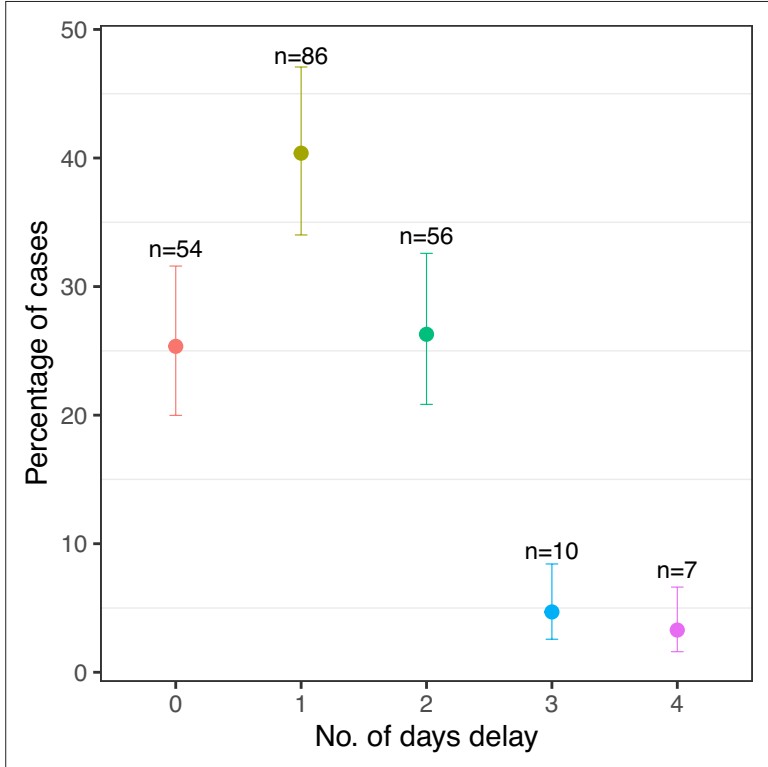

**Figure 2.** The number of days between abortion and the investigation (95% error bars, n = 213). No cases were investigated more than after 4 days after the abortion.

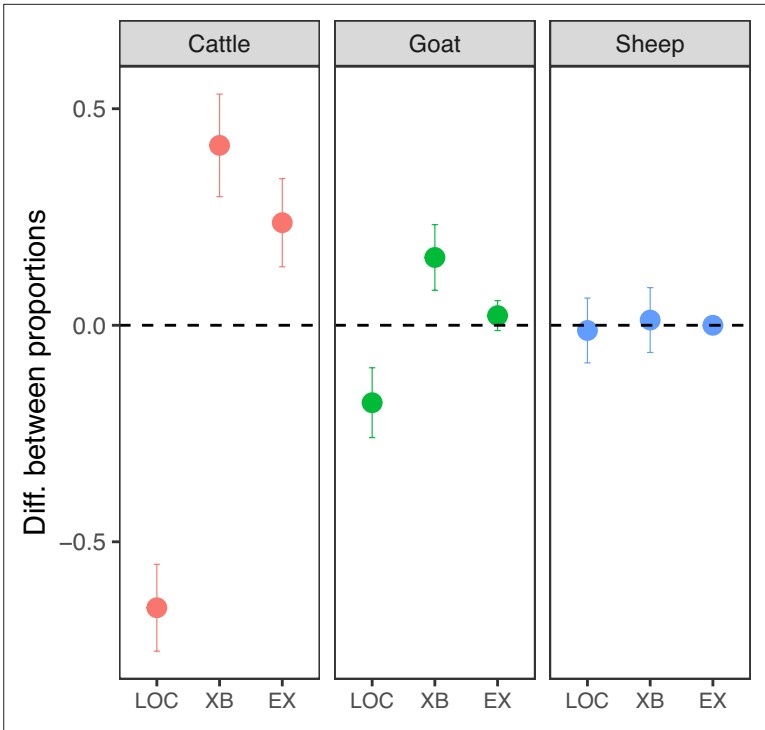

**Figure 3.** The difference between the expected and actual proportion of abortion cases in each species and breed (95% error bars, n = 215, exact binomial test, level of significance 0.05). Value of 0 = the expected number of cases occurred, >0 more than expected, <0 less than expected (LOC = indigenous (local), XB = non-indigenous cross-bred, EX = non-indigenous exotic breed).

From the Pearson correlation analysis, no significant association was found between the interval between reporting and investigation and the direct distance between the centroid of the ward in which the LFO worked and the research team headquarters ($R^2$=–0.07, $t$=–0.32, p=0.75) (*Figure 1b*).

## Sample collection data

Out of the 215 cases, placental and foetal tissues were collected in 116 (24.0%) and 141 (34.1%), respectively. The reasons given for failure to collect placental and foetal tissues included: (a) the tissues not being seen by owners (for example if the animal aborted while away grazing); (b) the tissues being burned by the owner; and (c) the tissues being consumed by dogs or other animals. Vaginal and milk samples from aborting dams were collected in 213 (99.1%) and 167 (77.7%) cases, respectively.

## Observed patterns in investigated abortions

### Pattern of abortions in species and breeds

In cattle, reported abortions occurred significantly more often than expected in non-indigenous cross-bred animals (expected proportion = 0.11, actual proportion = 0.52, 95% CI: 0.42–1.00, p<0.001) and non-indigenous exotic animals (expected proportion = 0.01, actual proportion = 0.25, 95% CI: 0.16–1.00, p<0.001). In goats, reported abortions occurred significantly more often than expected in non-indigenous cross-bred animals (expected proportion = 0.02, actual proportion = 0.18, 95% CI: 0.11–1.00, p<0.001) and more often than expected in non-indigenous exotic animals, although this difference was not significant (expected proportion = 0.01, actual proportion = 0.03, 95% CI: 0.01–1.00, p=0.053; *Figure 3* and *Supplementary file 2*). There was no significant difference in the distribution of abortions in different breeds of sheep.

### History of previous abortion cases in the aborting dams

Of the cattle, goat and sheep dams that were investigated in this study (and that had had previous pregnancies), 33.3% (n=12), 29.8% (n=17) and 16.7% (n=5), respectively, were reported by the owner

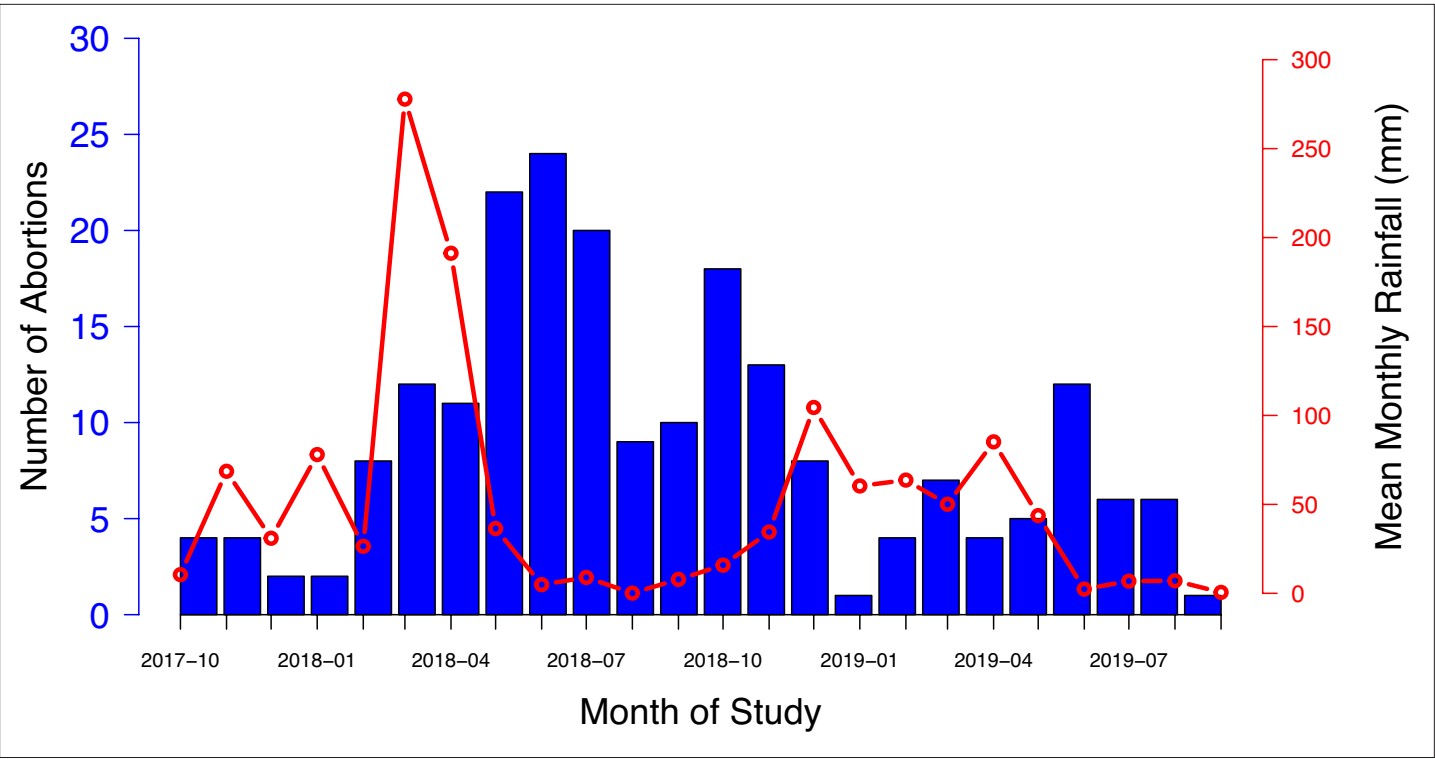

**Figure 4.** The number of abortion cases investigated per month (blue columns) shown against mean rainfall recorded in the Arusha region over each month of the study period (red line).

to have experienced a previous abortion event. Of these cattle and goats, 41.7% (n=5) and 47.1% (n=8), respectively, had suffered multiple previous abortions. In one particular case, a cow had experienced four previous abortion cases and one goat had experienced seven.

### History of recent stressful events

Of dams that aborted, 16 of 71 cattle (22.5%), 37 of 98 goats (37.7%) and 15 of 43 sheep (34.9%) were reported to have experienced recent stress. Dams that aborted for which an attribution was made were no more or less likely to have experienced a stressful event than dams for which an aetiological attribution was not made. Regarding recent illnesses over the previous four weeks (diagnosed by the farmer), cattle were reported to have suffered from a range of conditions including anaplasmosis, diarrhoea, lumpy skin disease, and trypanosomiasis, whilst goats were reported to have suffered predominantly from respiratory disease.

### Seasonality of cases

Cases of abortion were reported in every month of the 24-month study period, and although these fluctuated over time, with more cases reported during the drier periods (*Figure 4*), there was no significant effect of mean monthly rainfall on the number of cases (correlation coefficient = –0.005. $t$ = –0.25, p = 0.8).

### Determinants of attribution

As described in *Thomas et al., 2022*, the number of cases for which an abortigenic agent was attributed was 42 out of 215 (19.5%). Out of these, an attribution was made using PCR in 41 cases. One event in a cow met the case definition for two pathogens (both BHV-1 through seroconversion and *Neospora* through PCR) and in this event both pathogens were attributed. The attribution of the single case that was not determined using PCR (BHV-1) was made by serology alone. The sample types that were collected in each of these 41 cases for which an attribution was made using PCR, and whether the samples returned a positive or negative result, are shown in *Figure 5*. The time period between the

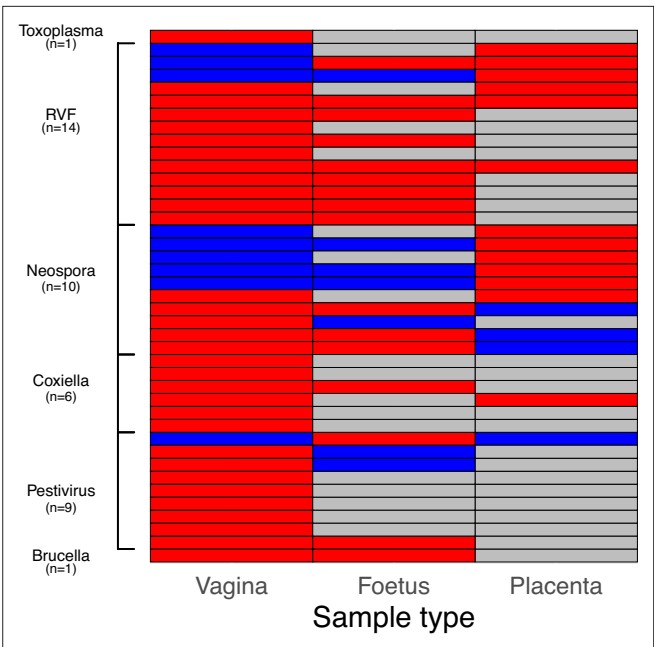

**Figure 5.** The type of samples in which pathogens were detected in the 41 abortion cases for which an attribution was made using PCR are shown. Each row represents one abortion event. Red - sample type returned a positive result; blue - sample type returned a negative result; grey - sample type was not collected.

abortion incident and the investigation (*Delay*) had a negative impact on attribution (z=–2.1, p=0.03), with each daily increase in the delay corresponding to a decrease in the odds of an attribution being made by 46.1% (i.e. 1–0.539; *Table 2* and *Figure 6*). Additionally, when the abortion occurred in goats (*Goat*) an attribution was significantly less likely than when it occurred in cattle. Finally, the odds of achieving an attribution were not affected by the availability of the placenta (*Placenta present*), foetus (*Foetus present*), or milk (*Milk collected*) for sampling.

## Exposure to zoonotic pathogens

Zoonotic pathogens (*Brucella* spp., *Coxiella burnetii*, *Toxoplasma gondii*. and RVFV) were detected in 61 of the 77 (79.2%) abortion cases where a pathogen was detected. Respondents reported that someone had assisted with the delivery in 13 (21.3%) of these cases, similar to the proportion of assisted deliveries across all abortion cases (40 out of 215, 23.5%). Of those assisting with delivery, the median age was 37, the youngest was seven, the eldest 84, and 20% were female.

**Table 2.** Output of final regression model investigating determinants of attribution.

|  | Odds Ratio | 2.5% | 97.5% | z value | p |
|---|---|---|---|---|---|
| (Intercept) | 0.353 | 0.053 | 2.361 | –1.074 | 0.283 |
| Delay | 0.539 | 0.306 | 0.95 | –2.139 | 0.032 |
| Goat | 0.195 | 0.058 | 0.653 | –2.651 | 0.008 |
| Sheep | 0.615 | 0.142 | 2.664 | –0.65 | 0.516 |
| Foetus present | 0.936 | 0.345 | 2.544 | –0.129 | 0.897 |
| Placental present | 1.779 | 0.621 | 5.094 | 1.074 | 0.283 |
| Milk collected | 2.405 | 0.639 | 9.055 | 1.297 | 0.195 |

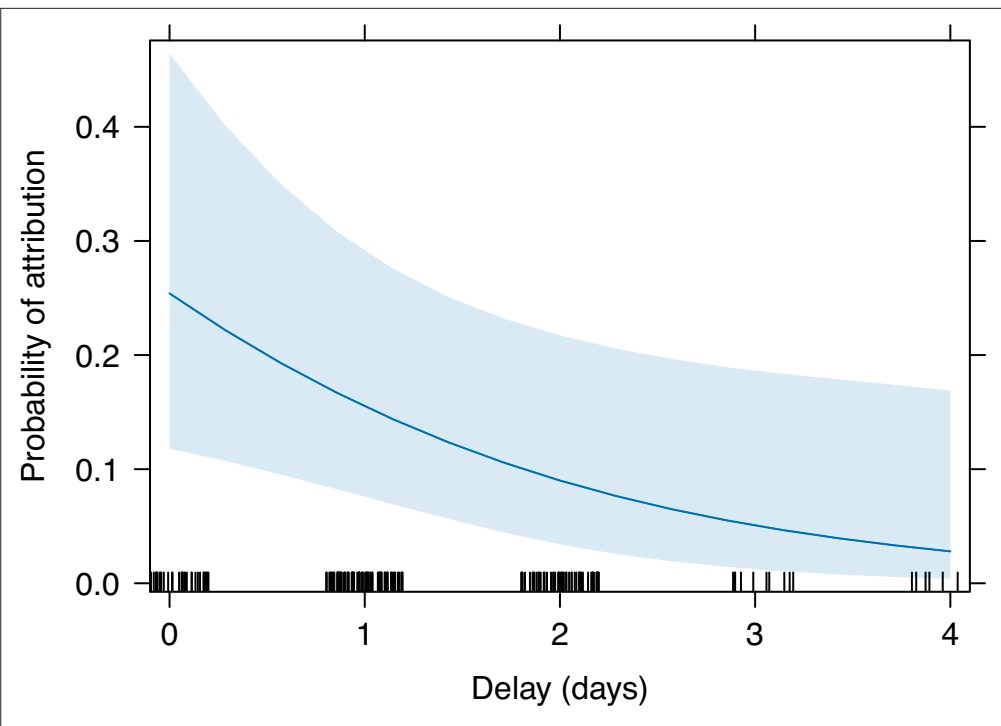

**Figure 6.** Predicted probability of attribution being made as a function of increasing delay between abortion and case investigation, as determined by the regression model output. The blue line indicates the regression line with the 95% confidence interval shaded blue.

## Discussion

This study demonstrated the feasibility of establishing a surveillance platform for reporting and diagnosing cases of livestock abortion and that these investigations, even at small scale, have generated important information on livestock diseases, and their investigation, in Tanzania. This included information that, to our knowledge, has not been captured or reported through existing surveillance systems.

Key findings were that: (i) livestock abortion can be a target for syndromic and/or event-based surveillance, with abortion incidents sufficiently distinctive and noteworthy to be reported by farmers to livestock field officers in a timely manner for investigation; (ii) pragmatic and robust protocols for sample collection and laboratory diagnosis can be established, even in resource-limited settings, to generate important etiological and epidemiological data; (iii) the likelihood of obtaining an etiological diagnosis depended on the timeliness of reporting and quality of sample collection; (iv) there was wide variation in reporting and investigation of cases by different LFOs and this did not appear to be associated with the distance between the ward and the investigation center; (v) in many herds prior livestock abortions were reported to have occurred in the 12 months preceding the investigated abortion event; (vi) abortions and repeat abortions reported disproportionately more frequently in non-indigenous breeds (cross-bred and exotic) than local breeds; and (vii) abortion cases wthere reported and investigated more frequently during drier periods, although there was no evidence of a relationship with monthly rainfall.

This study suggests that livestock abortion cases are sufficiently distinctive and observable to be reported by farmers to LFOs, would have widespread acceptability as a target for syndromic or event-based surveillance and would have considerable utility for farmers as well as for the livestock and public health sectors. The platform demonstrated the potential for generating high-quality aetiological data on previously relatively unrecognised livestock disease problems in Tanzania, such as *Neospora caninum* and novel pestiviruses, as well as data on endemic zoonoses, such as Q fever, brucellosis and toxoplasmosis (*Thomas et al., 2022*). Reports of unusual livestock abortion cases also provided an early warning of acute human health risks, as shown in this study area by the detection

of an outbreak of RVF in livestock in peri-urban smallholder cattle (*de Glanville et al., 2022*), which was the catalyst for subsequent public health investigations that identified several linked human RVF cases, including one fatality (*Madut et al., 2024*).

An attempt was made to gauge the sensitivity of the surveillance platform by estimating the percentage of expected abortions that the investigated cases represented. This was carried out using data on ward-level livestock numbers drawn from Ministry of Livestock and Fisheries records. However, given the dynamic nature of livestock populations in the study area and shifts in ward configurations, these estimates should be considered approximate figures only. Overall the sensitivity of this platform was low, with the highest ward level sensitivity (12.4%) below a mean sensitivity of 34% reported for bovine abortion surveillance systems in France (*Bronner et al., 2015*). Nonetheless, given that this was the first time that an abortion surveillance platform of this kind had been introduced, and the logistic and communication challenges inherent to this type of low-income setting, our study gives confidence of the feasibility and potential utility of this approach. Despite the limitations of the data, our conclusion that there is wide variation in the sensitivity of the surveillance platform across wards is likely to be robust. This finding, together with the ability of the platform to investigate only a very small proportion of expected abortion cases overall, has implications for the representativeness of data generated, which needs to be considered when interpreting the results reported here.

Contrary to expectations, the location and distance of the ward was not associated with the probability of cases being investigated. Association between cases investigated and farming system could not be determined as the number of wards was relatively small and the number representative of each system were not equal. However, relatively few cases were reported and investigated in the three agropastoral system wards, whilst both high and low rates of reporting occurred in individual wards within the smallholder (seven wards) and pastoral (five wards) systems. This suggests that LFOs and farmers in some wards were much more engaged and motivated than in others. To date, much investment in animal and human surveillance has been directed towards strengthening laboratory diagnostic capacity. While this is clearly essential, support also needs to be given to engaging, training, and motivating farmers and front-line animal health workers to effectively report both routine and unusual animal disease events. Understanding why and how animal keepers and LFOs are incentivised to report and investigate livestock health events will be essential for improving the reach and sensitivity of future surveillance platforms. The infrastructure for LFO engagement across Tanzania provides a valuable platform for timely and cost-effective reporting of abortions and other disease events but seems to be underutilised in existing surveillance systems.

The effectiveness of the platform in allowing an aetiological diagnosis to be reached was significantly impacted by the timeliness of investigation. The impact of delays may be explained by degradation of diagnostic samples, which also affected the utility of histopathological diagnosis (*Thomas et al., 2022*). Delays were caused by several factors, including the time taken by the investigation team to reach households due to remoteness of some areas and multiple cases being reported in different areas on the same day. These findings underscore the importance of being able to respond rapidly to cases and, where access is difficult, of providing locally suitable means of transport.

A reactive surveillance platform that is managed locally, for example by ward-based LFOs, will require the establishment of effective and safe protocols for collection and transport of samples for laboratory diagnosis that can be carried out without the need for long-distance travel of more highly trained investigators. Protocols for collection of vaginal swabs from aborting dams, which proved effective for both pathogen detection and attribution, may be of value. In this study, the greater accessibility of vaginal swabs, as compared to placental samples and foetal samples, meant more attribution of abortigenic pathogens was attained through these samples (*Figure 5*). They also require less handling of potentially infectious tissues and are more reliably accessible than placental and foetal tissues, which are often consumed by scavengers or disposed of by the farmer. Indeed, collection of vaginal swabs from the dam was possible in almost all cases investigated in this study. Thus, while logistic, financial, and capacity constraints for comprehensive sampling and investigation of livestock abortions are likely to exist across Africa, these need not preclude the establishment of simple and robust protocols that can yield valuable surveillance data.

## Attribution

From this platform, an attribution was reached in 19.5% of cases (*Thomas et al., 2022*), which was not far below the typical range of 25–45% achieved in industrialized farming systems in high-income settings (*Campero et al., 2003*; *Amouei et al., 2019*; *Derdour et al., 2017*; *Anderson et al., 1990*). During this study, samples were tested for only 10 abortigenic pathogens and it is likely that rates of detection and attribution would be higher with inclusion of tests for other known abortigenic agents (such as *Listeria* spp., *Campylobacter* spp., *Salmonella* spp. and fungal pathogens), with more locally relevant aetiological data informing the suite of pathogens to be tested, and with metagenomic approaches.

As elsewhere in the world, non-infectious conditions, including nutritional, metabolic, and toxic conditions (*Alemayehu et al., 2021*; *Mee et al., 2023*; *Woodburn et al., 2021*), are also likely to cause livestock abortions in Tanzania. Our study suggests a possible role for recent stressful events, including episodes of drought and attacks by wild predators, which were reported from approximately a third of unattributed abortion cases. However, these events are a common occurrence in our study population, so attribution is likely to be challenging.

We do not have an immediate explanation as to why the likelihood of attribution was lower for goats than for sheep or cattle. One possible explanation is that, in comparison to sheep and cattle, fewer data are available worldwide on causes of abortions in goats and very little is known about the aetiology of goat abortions in Africa, so the pathogens included in our diagnostic panel may not have been as relevant for goats as for sheep and cattle. This lack of knowledge is highlighted by the unusual finding of *Neospora caninum* as a cause of abortion in a goat detected through this platform (*Thomas et al., 2022*).

The challenges of attribution, discussed also in *Thomas et al., 2022*, should not detract from the value of reporting and investigating livestock abortion cases. Recent studies have indicated that the economic costs of livestock abortion in Tanzania and impacts on food security are much more substantial than previously recognized; for example (*Semango et al., 2024*) estimated the annual gross losses associated with abortion in Tanzania to be $262 million USD. Given these findings, data collected on the number of cases and species/breed affected will be of considerable value in highlighting the importance of this syndrome in the context of livestock productivity, household livelihoods and food security.

## Patterns of livestock abortion in northern Tanzania

Data from the household questionnaire indicated that livestock abortion was a common occurrence across the study area with almost a third of cattle herds and two-thirds of small ruminant herds having experienced a previous case of abortion in the preceding 12 months.

This study indicated that investigated cases occurred more often than expected in non-indigenous breeds (cross-bred and exotic cattle and cross-bred goats) than in local breeds. This is consistent with studies reporting a four-times higher risk of abortion in cross-bred cattle in Ethiopia in comparison to local breeds (*Deresa et al., 2020*) and studies in Nigeria (*Yakubu et al., 2015*) and India (*Khan et al., 2016*) reporting breed effects in relation to cattle abortion. Our results may simply reflect a higher likelihood of reporting of abortion in animals that are of higher value and more likely to be zero-grazed or tethered while grazing (and hence abortions would be more apparent) and for which an existing relationship with animal health services is more likely.

More than a fifth of the dams that aborted were reported to have experienced a previous abortion, and several had experienced multiple abortion losses, with non-indigenous breeds more likely than indigenous breeds to have had a previous abortion. Chronic repeat abortions can arise from infection with pathogens such as *Neospora caninum* and pestiviruses which were detected as common causes of abortion in this study. Repeat aborting dams could be identified as candidates for slaughter with potential benefits for herd productivity, as well as reduced risks of potential zoonotic transmission. However, any advice around removal of these animals from a herd may result in sale rather than slaughter, with the risk of further spread of the pathogen.

The risk of exposure to zoonotic pathogens through abortion cases was underscored by two findings: first that, in nearly 80% of cases in which a pathogen was detected, the pathogen was zoonotic and second, that in nearly a quarter of all cases, someone had assisted with the aborted delivery, very likely without any form of personal protective equipment. Of those who assisted, 20% were female, of

which most were of reproductive age and therefore at particular risk following infection with pathogens such as *Toxoplasma gondii.*

In conclusion, this study has demonstrated that livestock abortion surveillance, even at a relatively small scale, can capture valuable aetiological and epidemiological information on important livestock pathogens, including those that are zoonotic and those with epidemic potential. The study demonstrated the utility, acceptability, and feasibility of livestock abortion as a target for both syndromic and event-based surveillance and showed that an effective reporting and investigation system could be operationalized across a range of settings in Tanzania, including remote rural areas. Key elements of effectiveness were high levels of engagement of community-based field officers, the establishment of practical and robust field sample collection and application of molecular diagnostic techniques, with prompt response to reporting of cases, and timely feedback of results. Future research might include abortion surveillance implemented at a larger scale to better understand the extent, and economic impact, of livestock abortion and associations with different livestock keeping systems. This work might include the investigation of rapid point-of-care testing and the use of mobile phone technology to speed up the detection and reporting back of results to LFOs and livestock keepers, with the potential to facilitate the capture of data for use in national disease surveillance databases. Such research might allow for refined assessment of the economic impact of abortion and, subsequently, of preventative interventions. Further, our results suggest a possible link between abortion and low rainfall, and further work is needed to examine this potentially important relationship in detail. Additionally, investigating the effectiveness of syndromic or event-based surveillance more broadly, targeting syndromes other than abortion, would provide further evidence demonstrating whether this type of

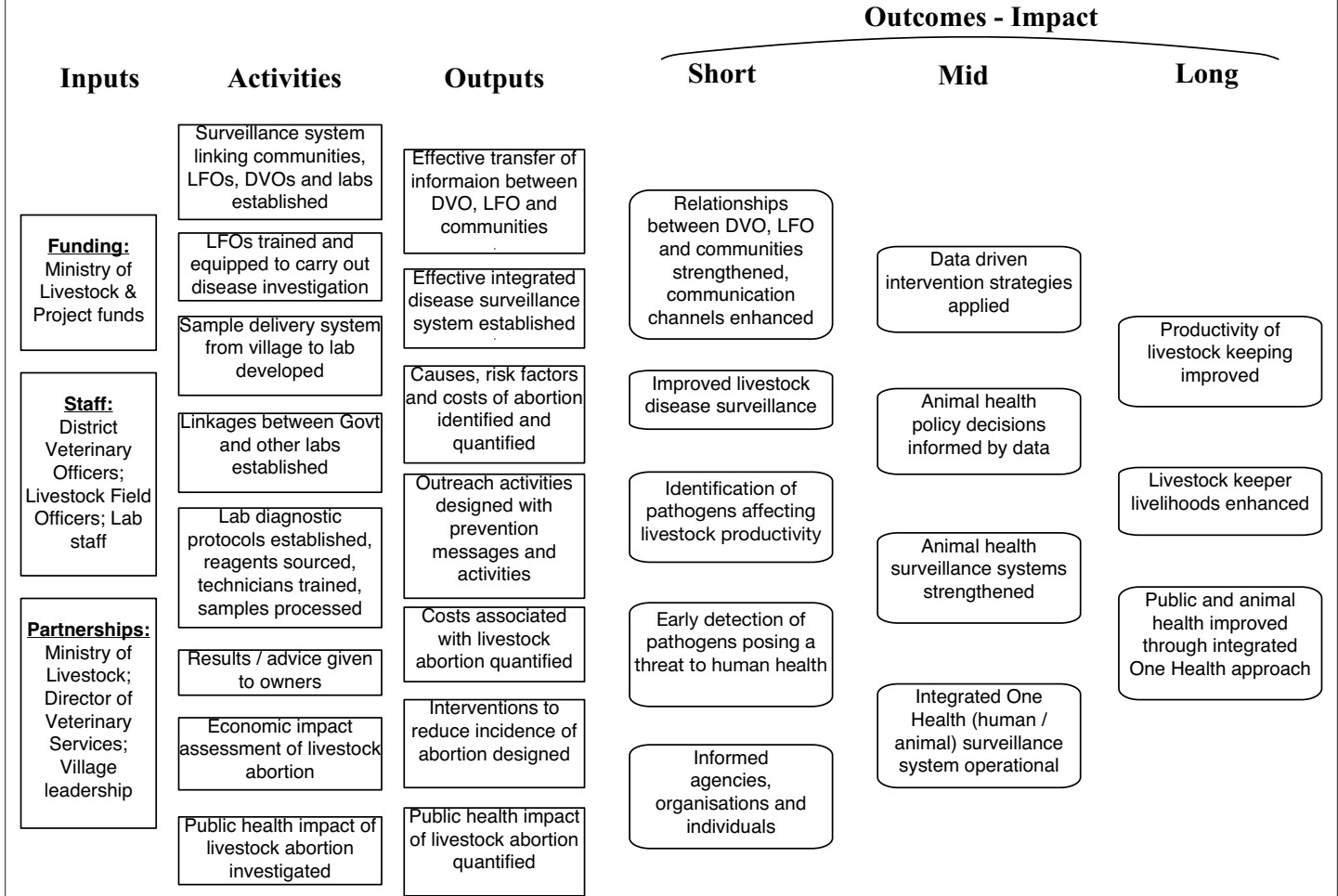

**Figure 7.** A logic model illustrating the conceptual links between the inputs, activities, outputs, and short to long-term impacts expected from effective livestock health surveillance with a particular focus on abortion.

surveillance can be implemented, in a cost-effective manner, at scale. Such research would provide much needed evidence to inform the design of effective livestock health interventions to improve livelihoods, food security, and public health.

## Materials and methods

### Logic model

A logic model was created to provide a conceptual framework that described the logical links between the main activities, outputs, and outcomes that were expected from the programme (*Figure 7*). The model depicts the overarching assumption that building better livestock abortion surveillance systems and strong community partnerships will lead to data-driven interventions to prevent and control infectious causes of livestock abortion and to catalyse changes in knowledge, attitudes, behaviours, or practices that could improve livestock productivity, livelihoods, and human and animal health.

### Abortion surveillance platform

The abortion surveillance platform was set up in northern Tanzania through a collaboration between the Ministry of Livestock and Fisheries, local government authorities, and the research team. The study was undertaken from October 2017 through September 2019 in 15 wards of five districts of Arusha, Kilimanjaro, and Manyara Regions in northern Tanzania (*Figure 8*). These study wards were selected from randomly selected wards included in earlier cross-sectional exposure studies (*Bodenham et al., 2021*). Thirteen wards were selected at random and two additional wards were selected purposively because of strong existing relationships with the livestock-keeping community (*Thomas et al., 2022*). These 15 wards comprised five wards that were expected to be predominantly pastoral, three were expected to be predominantly agropastoral and seven expected to predominantly smallholder, with categories assigned by the research team following discussion with local experts (typically the district level veterinary officer; *Bodenham et al., 2021*). Recruited livestock field officers (LFOs) responsible for each target ward received training on the causes and safe investigation of livestock abortion. These officers are government employees that are equivalent to para-veterinarians in other settings. They were requested to ask livestock owners to report any incidents of livestock abortion, stillbirths, and peri-natal death (hereafter referred to as abortion cases).

### Investigation of cases

Cases were investigated if, following a report from the LFO, the event could be followed up within 72 hr of the abortion occurring. Full details of sample collection are provided in *Thomas et al., 2022*. Briefly, where available, blood, milk, and vaginal swabs were collected from the aborting dam and tissue and swab samples collected from the foetus and placental membranes. Information about the abortion event was collected and a household questionnaire (comprised of mixed open and closed questions) conducted to collect information on livestock demographics, livestock abortion history, the aborting dam (age, breed), household livestock parturition practices and household socio-economic data (*Supplementary file 3*). Questionnaire data were only collected from abortion cases that were investigated by the research team and were used to investigate underlying patterns of abortion, risks associated with abortion cases, and operational aspects of the surveillance platform.

LFOs were instructed to provide farmers with advice as to locally appropriate preventive measures that could be taken to reduce transmission or contamination risks associated with abortion cases, which included safe removal of abortion tissues from livestock-occupied areas (e.g. burning, burying or covering the tissues in thorny branches; *Supplementary file 4*). Diagnostic results were reported back to LFOs and livestock owners within 10 days of the investigation and, where pathogens were detected, more specific advice provided as to appropriate management strategies that could minimise further transmission to livestock and people.

Event data were collected using a paper-based Cardiff Teleform system (Cardiff Inc, Vista, Ca., USA) into an Access database (Microsoft Corp, Va., USA). Household questionnaire data were collected using handheld digital devices programmed with the Open Data Kit survey tool. Data were imported into R (*R Development Core Team, 2023*) for cleaning, coding and analysis. The survey instruments were pre-tested in wards that were not targets for this study. Geographic co-ordinates from a central point within the household were collected with a handheld GPS (Garmin eTrex).

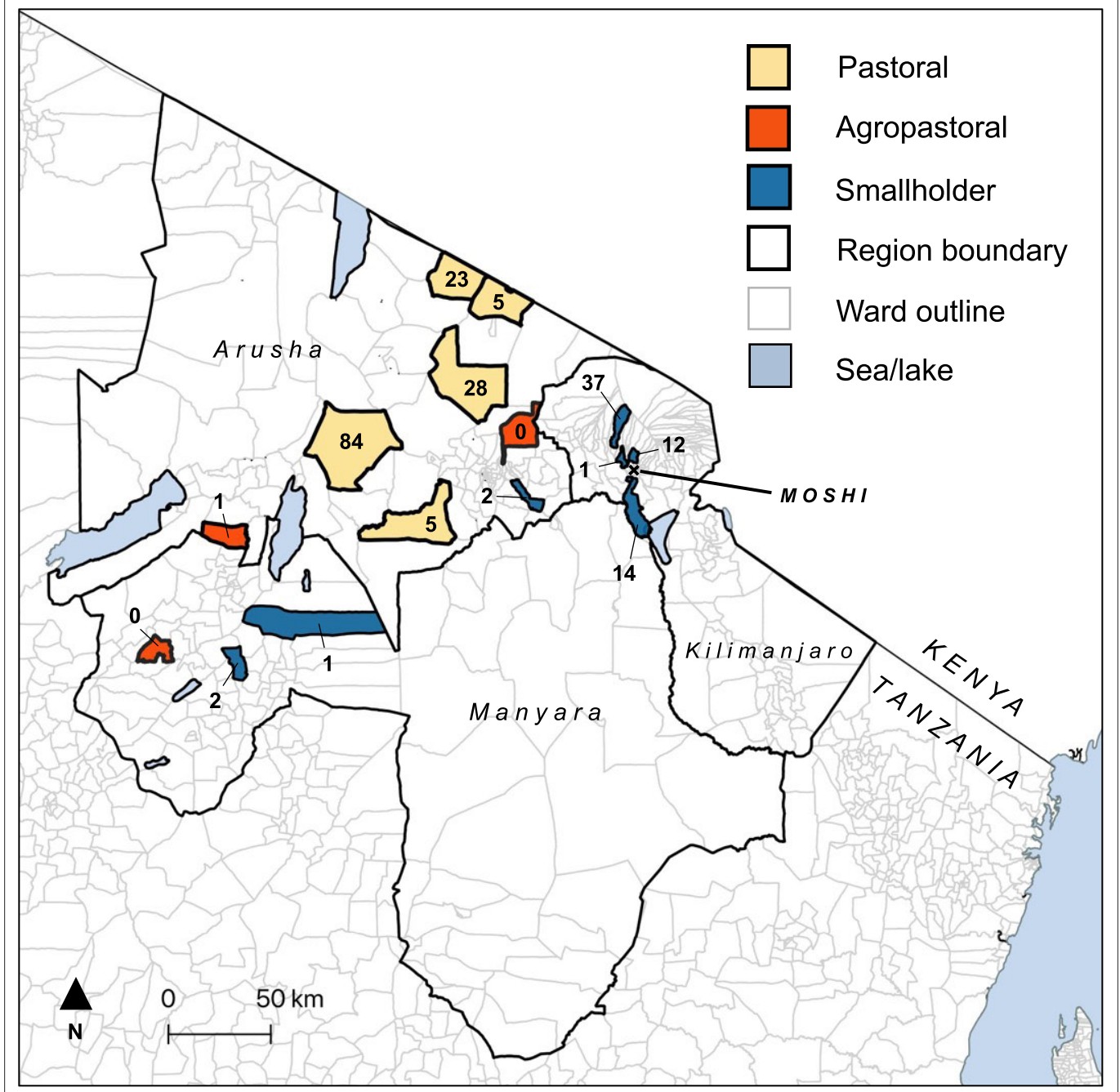

**Figure 8.** Map of the study area in northern Tanzania showing selected pastoral, agropastoral and smallholder wards in Kilimanjaro, Arusha, and Manyara Regions. The number of investigated cases per ward and the study base (Moshi) are shown.

## Sample analysis

Laboratory diagnostic analyses have previously been described in detail (*Thomas et al., 2022*). Briefly, samples were tested for: (a) *Brucella* spp., *Coxiella burnetii*, *Chlamydia* spp., and *Leptospira* spp. using quantitative polymerase chain reaction (qPCR) assays: (b) *Neospora* spp. and *Toxoplasma gondii* using conventional nested PCR assays; (c) Bluetongue virus (BTV), bovine viral diarrhoea virus / border disease virus (BVDV/BDV), and Rift Valley fever virus (RVFV) using quantitative reverse transcriptase PCR (RT-qPCR); and (d) additional tests for *C. burnetii* and *Neospora* spp. using immunohistochemistry. Additionally, serum was tested for antibodies to *Brucella* spp., bovine herpesvirus (BHV-1), BVDV, *C. burnetii*, *Leptospira hardjo*, and RVFV.

## Summarising the investigated cases and description of the livestock study population

The number of abortion cases was recorded for each agro-ecological zone (pastoral, agro-pastoral and small holder) and study ward, together with the number of households that had an abortion event, the composition of the livestock herds (cattle, goats, and sheep) kept at each household, and the number of previous abortion cases in each household that had an event.

The sensitivity of the platform was examined by dividing the number of investigated abortion cases by the *expected* number of abortions for the livestock population of the study wards over the study period. The expected number of abortions was estimated by multiplying (a) the number of abortion cases per head of livestock reported over a 12-month period obtained through a previous randomized cross-sectional study (described in *de Glanville et al., 2022*; *Supplementary file 5*) by (b) the number of livestock in the ward reported by surveys conducted by LFOs from the Tanzania Ministry of Livestock and Fisheries covering a period from 2011 to 2016 (E. Swai, *unpublished data*; *Supplementary file 6*). This figure was multiplied by two to account for the 24-month duration of the study. Where data were not available for a specific ward, figures were estimated from the average of other wards in the same division (Rau) or by current estimates provided by the LFO (Machame Mashariki).

## Determinants of investigation

The determinants of event investigation were analysed to determine the number of cases that each LFO investigated and the distribution of time taken between the report and the subsequent investigation by the research team. The relationship between the time taken between a report and its subsequent investigation and the distance (km) of the ward from the research team headquarters (in the town of Moshi) was investigated using Pearson correlation analysis (*Figure 1*).

## Sample collection data

Data were recorded and summarized for: (a) the types of samples collected for each abortion event; (b) the availability of placental and foetal materials and, when not available, the reasons why these materials were not available; and (c) the relationship between sample types and pathogen detection and pathogen attribution.

## Observed patterns in investigated abortions

The distribution of investigated abortion cases was examined in relation to (a) livestock species and breed in which they occurred, (b) history of previous abortion cases in the aborting dams, (c) a history of recent stressful events affecting the pregnant dam, and (d) seasonality of the cases.

a. Breeds were recorded as indigenous (local) and non-indigenous (cross-bred and exotic). We used the breed distribution in the investigated herds to estimate the expected number of abortions if breed type did not affect the likelihood of abortion. Using an exact binomial test, we compared the expected with the observed number of cases.
b. Data on the occurrence of previous abortion cases and recent stressful incidents were obtained from the questionnaire data collected at the time of the investigation.
c. Stressful incidents were recorded as open-ended questions but were prompted by asking about events such as being chased by predators, drought, unusual handling or recent change in diet or grazing habit.
d. Seasonality was investigated in relation to typical 'dry' and 'wet' seasons and in relation to monthly rainfall in the Arusha region during the project period. Mean monthly rainfall was calculated from daily rainfall data, which was obtained across the Arusha Region (*Ashouri et al., 2015*; *Soroosh et al., 2019*) and plotted alongside the temporal pattern of reported cases. The relationship between mean monthly rainfall and the number of reported abortions was investigated using linear regression.

   All processing and analysis of data was carried out using the statistical programming tool, R (*R Development Core Team, 2023*).

## Determinants of attribution

World Organization for Animal Health (WOAH – previously OIE) guidelines and case definitions, in conjunction with recommendations from specialist/reference laboratories and peer-reviewed literature, were used to inform diagnosis and attribution of infectious causes of abortion (described in detail in *Thomas et al., 2022*). Binomial logistic regression analysis was carried out with attribution (yes or no) as the dependent variable. Potential independent explanatory variables, including the delay between the event and its investigation, species, and livestock management system, were selected for inclusion in the model through univariable analysis (with all variables with p-value < 0.25 selected for inclusion). Potential independent explanatory variables were retained or de-selected through a stepwise approach using Akaike Information Criterion (AIC) as an indicator for model efficiency to achieve a final model.

## Exposure to zoonotic pathogens

The percentage of cases in which zoonotic pathogens (*Brucella* spp., *C. burnetii*, *T. gondii* and RVFV) were detected, and the percentage of these cases in which someone assisted with the delivery, was calculated. The age and sex of the persons assisting with delivery was also collected.

## Acknowledgements

We are grateful for the support of the Ministry of Livestock and Fisheries, the District Veterinary Officers, the Livestock Field Officers and the communities that contributed to this work. We are also grateful to the research team who implemented the project.

## Additional information

### Funding

| Funder | Grant reference number | Author |
|---|---|---|
| University of Edinburgh | R83537 | Sarah Cleaveland |
| Biotechnology and Biological Sciences Research Council | BB/L018926/1 | Sarah Cleaveland |

The funders had no role in study design, data collection and interpretation, or the decision to submit the work for publication.

### Author contributions

Felix Lankester, Conceptualization, Data curation, Formal analysis, Supervision, Investigation, Methodology, Writing – original draft, Project administration, Writing – review and editing; Tito J Kibona, Nick Wheelhouse, Emanuel Swai, Project administration, Writing – review and editing; Kathryn J Allan, Conceptualization, Data curation, Formal analysis, Writing – review and editing; William de Glanville, Validation, Writing – review and editing; Joram J Buza, Obed M Nyasebwa, Supervision, Project administration, Writing – review and editing; Frank Katzer, Investigation, Methodology, Writing – review and editing; Jo E Halliday, Writing – review and editing; Blandina T Mmbaga, Supervision, Methodology, Project administration, Writing – review and editing; Elisabeth A Innes, Data curation, Writing – review and editing; Kate M Thomas, Data curation, Formal analysis, Investigation, Methodology, Project administration, Writing – review and editing; John R Claxton, Conceptualization, Funding acquisition, Project administration, Writing – review and editing; Sarah Cleaveland, Conceptualization, Data curation, Formal analysis, Funding acquisition, Investigation, Methodology, Writing – original draft, Project administration, Writing – review and editing

### Author ORCIDs

Felix Lankester ⬤ https://orcid.org/0000-0002-0802-0693
Kathryn J Allan ⬤ https://orcid.org/0000-0001-6612-889X
Frank Katzer ⬤ https://orcid.org/0000-0001-5902-4136
Jo E Halliday ⬤ https://orcid.org/0000-0002-1329-9035

Kate M Thomas ⓘ https://orcid.org/0000-0002-1589-8314

### Ethics

Ethical ApprovalEthics approval for this research was granted by Kilimanjaro Christian Medical Centre (KCMC) Ethics Committee (No, 535 and No. 832); National Institute of Medical Research (NIMR), Tanzania (NIMR/HQ/R8a/Vol IX/1522 & NIMR/HQ/R.8a/Vol IX.2028); Research Ethics Coordination Committee and the Institutional Review Board for Clinical Investigations of Duke University Health System in the United States (Pr00037356); University of Otago Ethics Committee (H15/069 & H17/069); University of Glasgow College of Medical Veterinary and Life Sciences Ethics Committee (200140152 & 200170006).

Reviewer #1 (Public review): https://doi.org/10.7554/eLife.95296.3.sa1
Reviewer #2 (Public review): https://doi.org/10.7554/eLife.95296.3.sa2
Reviewer #3 (Public review): https://doi.org/10.7554/eLife.95296.3.sa3
Author response https://doi.org/10.7554/eLife.95296.3.sa4

## Additional files

### Supplementary files

• Supplementary file 1. Herd level summary statistics are provided.

• Supplementary file 2. The actual number of abortions reported for each species and breed and, based on the proportion of each breed in all the herds that reported cases, the expected number of abortions.

• Supplementary file 3. A copy of the household questionnaire (comprised of mixed open and closed questions) conducted to collect information on livestock demographics, livestock abortion history, the aborting dam (age, breed), household livestock parturition practices and household socio-economic data.

• Supplementary file 4. Instructions to provide farmers with advice as to locally appropriate preventive measures that could be taken to reduce transmission or contamination risks associated with abortion cases, which included safe removal of abortion tissues from livestock-occupied areas (e.g. burning, burying or covering the tissues in thorny branches).

• Supplementary file 5. The number of abortion cases per head of livestock reported over a 12 month period obtained through a previous randomized cross-sectional study (described in *de Glanville et al., 2022*).

• Supplementary file 6. The number of livestock in the ward reported by surveys conducted by Livestock Field Officers from the Tanzania Ministry of Livestock and Fisheries covering a period from 2011–2016 (E. Swai, *unpublished data*).

• MDAR checklist

### Data availability

All data generated or analysed during this study are included in the manuscript and supporting files.

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
