## [Editor Report · eLife assessment]

This **important** study reports the use of a surveillance approach in identifying emerging diseases, monitoring disease trends, and informing evidence-based interventions in the control and prevention of livestock abortions, as it relates to their public health implications. The data support the **convincing** finding that abortion incidence is higher during the dry season, and occurs more in cross-bred and exotic livestock breeds. Aetiological and epidemiological data can be generated through established protocols for sample collection and laboratory diagnosis. These findings are of potential interest to the fields of veterinary medicine, public health, and epidemiology.

---

## [Referee Report · Reviewer #1 (Public review)]

Summary:

The paper examined livestock abortion, as it is an important disease syndrome that affects productivity and livestock economies. If livestock abortion remains unexamined it poses risks to public health.

Several pathogens are associated with livestock abortions but across Africa however the livestock disease surveillance data rarely include information from abortion events, little is known about the aetiology and impacts of livestock abortions, and data are not available to inform prioritisation of disease interventions. Therefore the current study seeks to examine the issue in detail and proposes some solutions.

The study took place in 15 wards in northern Tanzania spanning pastoral, agropastoral and smallholder agro-ecological systems. The key objective is to investigate the causes and impacts of livestock abortion.

The data collection system was set up such that farmers reported abortion cases to the field officers of the Ministry of Livestock and Fisheries livestock

The reports were made to the investigation teams. The team only included abortion of those that the livestock field officers could attend to within 72 hours of the event occurring.

Also a field investigation was carried out to collect diagnostic samples from aborted materials. In addition aborting dams and questionnaires were administer to collect data on herd/flock management. Laboratory diagnostic tests were carried out for a range of abortigenic pathogens

Over the period of the study 215 abortion events in cattle (n=71), sheep (n=44) and goats (n=100) were investigated. In all 49 investigated cases varied widely across wards, with three .The Aetiological attribution, achieved for 19.5% of cases through PCR-based diagnostics, was significantly affected by delays in field investigation.

The result also revealed that vaginal swabs from aborting dams provided a practical and sensitive source of diagnostic material for pathogen detection.

Livestock abortion surveillance can generate valuable information on causes of zoonotic disease outbreaks, and livestock reproductive losses and can identify important pathogens that are not easily captured through other forms of livestock disease surveillance. The study demonstrated the feasibility of establishing an effective reporting and investigation system that could be implemented across a range of settings, including remote rural areas,

Strengths:

The paper combines both science and socio economic methodology to achieve the aim of the study.

The methodology was well presented and the sequence was great. The authors explain where and how the data was collected. Figure 2 was used to describe the study area which was excellently done. The section on Investigation of cases was well written. The sample analysis was also well written. The authors devoted a section to summarizing the investigated cases and description of the livestock 221-study population. The logic model has been well presented

Weaknesses:

All the weaknesses identified have been resolved by the the authors

---

## [Referee Report · Reviewer #2 (Public review)]

The paper provides a comprehensive analysis of the importance of livestock abortion surveillance in Tanzania. The authors aim to highlight the significance of this surveillance system in identifying disease priorities and guiding interventions to mitigate the impact of livestock abortions on both animal and human health.

Summary:

The paper begins by discussing the context of livestock farming in Tanzania and the significant economic and social impact of livestock abortions. The authors then present a detailed overview of the livestock abortion surveillance system in Tanzania, including its objectives, methods, and data collection process. They analyze the data collected from this surveillance system over a specific period to identify the major causes of livestock abortions and assess their public health implications.

Evaluation:

Overall, this paper provides valuable insights into the importance of livestock abortion surveillance as a tool for disease prioritization and intervention planning in Tanzania. The authors effectively demonstrate the utility of this surveillance system in identifying emerging diseases, monitoring disease trends, and informing evidence-based interventions to control and prevent livestock abortions.

Strengths:

(1) Clear Objective: The paper clearly articulates its objective of highlighting the value of livestock abortion surveillance in Tanzania.

(2) Comprehensive Analysis: The authors provide a thorough analysis of the surveillance system, including its methodology, data collection process, and findings as seen in the supplementary files.

(3) Practical Implications: The paper discusses the practical implications of the surveillance system for disease control and public health interventions in Tanzania.

(4) Well-Structured: The paper is well-organized, with clear sections and subheadings that facilitate understanding and navigation.

All suggestions made for improvement of the manuscript have been appropriately effected.

Final Recommendation:

Overall, this paper makes a significant contribution to the literature on livestock abortion surveillance and its implications for disease control in Tanzania.

---

## [Referee Report · Reviewer #3 (Public review)]

The authors delved into an important aspect of abortifacient diseases of livestock in Tanzania. The thoughts of the authors on the topic and its significance have been clarified. The number of wards in the study area, statistical selection of wards, type of questionnaire ie open or close ended. and statistical analyses of outcomes have been clearly elucidated in the manuscript. The exclusion criteria for two wards out of the fifteen wards mentioned in the text are clearly stated. Observations were from pastoral, agro-pastoral and small holder agro ecological farmers. Sample numbers or questionnaires attributed to the above farming systems correlate findings with management systems. The impacts of the research investigation output are clearly visible as to warrant intervention methods. The identified pathogens from laboratory investigation, particularly with the use of culture and PCR, as well as the zoonotic pathogens encountered are stated in the manuscript and the supplementary files.

In conclusion, based on the intent of the authors and content of this research, and the weight of the research topic, the seeming weaknesses in the critical data analysis observed have been clarified, to demonstrate cause, effect and impact.

The authors have carried out the necessary corrections.

The findings do imply that identification of some of the abortifacient of livestock in Tanzania will necessitate important interventions in the control of the diseases in the study area

---

## [Author Response]

The following is the authors’ response to the original reviews.

**Reviewer 1:**
Lines 43 to 46 cannot be referred to as methodology:"to investigate (a) determinants of attribution; (b) patterns of investigated events, including species and breed affected, history of previous abortion and recent stressful events, and the seasonality of cases; (c) determinants of reporting, investigation and attribution; (d) cases in which zoonotic pathogens were detection".The above should be deleted from the methodology.

The text is in the abstract and describes, in brief, analyses that we performed and the rationale for these analyses, which we consider relevant for understanding the approach. As such, we think the text should remain.

Italicize et al. in the citations

This has been done.

**Reviewer 2:**
Data Presentation: While the analysis is comprehensive, the presentation of data could be enhanced with the use of more visual aids such as tables, graphs, or charts to illustrate key findings.

While further visualisation of findings would be possible, we consider the key results are captured effectively in the existing figures and tables. Open access to the data also allows for further analyses that might be of interest to readers.

Discussion Section: The paper could benefit from a more in-depth discussion of the implications of the findings for disease control strategies and policy formulation in Tanzania.

We thank the reviewer for this important comment. In most of the paragraphs of the Discussion we discuss the implications of the findings with specific reference, where relevant, to disease control in Tanzania. For example, in the paragraph regarding human capacity building, we discuss how LFOs might be incentivised to report health events and how this could improve the reach and sensitivity of future surveillance platforms. Similarly, these issues are discussed in other paragraphs of the Discussion.

Future Directions: Including recommendations for future research or areas for further investigation would add depth to the paper.

This suggestion has been acted upon and we have added text in the conclusion to describe recommendations for future research.

**Reviewer 3:**
The thoughts of the authors on the topic and its significance are implied, and the methodological approach needs further clarity. The number of wards in the study area, statistical selection of wards, type of questionnaire ie open or close-ended. Statistical analyses of outcomes were not clearly elucidated in the manuscript.

The number of wards and how they were selected (from randomly selected wards included in earlier cross-sectional exposure studies (Bodenham et al. 2021)) is described in the Abortion Surveillance Platform section of the Methods. We have added description of the questionnaire to indicate that it was a mixture of open and closed questions. We have reviewed the statistical analyses and consider that they have been fully and appropriately described and so have not changed this.

Fifteen wards were mentioned in the text but 13 used what were the exclusion criteria.

As described, the study focussed on fifteen wards however two wards did not report any cases. As such, investigations only took place in thirteen of the fifteen wards and this has been described in the text.

Observations were from pastoral, agropastoral, and smallholder agroecological farmers. No sample numbers or questionnaires were attributed to the above farming systems to correlate findings with management systems.

As described, the 15 wards comprised five wards that were expected to be predominantly pastoral, three were expected to be predominantly agropastoral and seven expected to predominantly smallholder, and these categories were assigned by the research team following discussion with local experts (typically the district level veterinary officer) (Bodenham et al. 2021). As such, we consider this to be described sufficiently.

The impacts of the research investigation output are not clearly visible as to warrant intervention methods.

The aim of this paper was to provide insights on the feasibility and value of establishing a livestock abortion surveillance platform. The aetiological data that could be used to inform specific disease control measures or interventions was the focus of a previous paper (Thomas et al. 2022) as described in the text.

What were the identified pathogens from laboratory investigation, particularly with the use of culture and PCR not even mentioning the zoonotic pathogens encountered if any?

An earlier published paper describing the aetiology of the cases was mentioned (Thomas et al. 2022). This paper fully describes the identified pathogens and the methods used for identification and attribution. Additionally, in the Sample Analysis section we describe the pathogens that were tested and the methods used. In the section Exposure to Zoonotic Pathogens we specifically list Brucella spp. C. burnetiid, *T. gondii* and RVFV and so we consider that we have sufficiently described the pathogens tested for, the methods and the zoonotic pathogens detected.

The public health importance of any of the abortifacient agents was not highlighted.

The Introduction provides background information on the public health importance of abortifacient agents and we dedicate a whole section (Exposure to Zoonotic Pathogens) to the public health implications of the number of cases in which zoonotic pathogens were detected. Additionally, we discuss the implications of this in the Discussion.

Comments in manuscript itself:Line 230: Why are you estimating. The study was supposed to be based on real time abortion events or at least abortion events within 72 hours

We were estimating the sensitivity of the platform by dividing the number of investigated abortion cases by the number of abortions for the livestock population in each of the study wards that would have been expected over the study period. Because the denominator in this calculation was an expected number, and not a measured count, we can only estimate.

236: In areas where there was no reported abortion event why will you estimate. This action will lead to false conclusion of abortion event in area that did have an event.

We think there has been some misunderstanding of what this section of text was describing. We were not attributing a case to an area where there was none. Rather, as mentioned above, the aim of this particular analysis was to estimate the sensitivity of the platform. To achieve this, we needed to estimate what the expected number of abortion cases in each ward would have been.

279: Give a brief description of R

A citation and some explanatory text have been added.

348: Table 1: Your table did not show cases where estimate values were used

We think this comment has resulted from the confusion described above regarding estimated cases. Table 1 has summary data for the actual cases that were reported in the study and does not have the data for the estimated number of abortions that were expected to have occurred in each ward. As described in line 247, this data is given in Supplementary Materials 3.

404: Not clear, please rephase

This sentence has been re-drafted to improve clarity

467: Why are you numbering the findings of your investigation in your discussion? You have not told us about the previous abortion event in your study area prior to this study and why you embarked on this study in this regions. The current abortion event situation in your country based on other researchers work is missing and how your findings is important as it related to similar investigation elsewhere.

We number the key findings for clarity and to make each finding distinct and so prefer to retain it.

The study area was chosen because it was the site of an earlier cross-sectional exposure study within which the wards were randomly selected. As a result, thirteen of the fifteen wards targeted in the reported study were randomly selected. Two additional wards were selected purposively because of strong existing relationships with the livestock-keeping community. This was explained in the Methods in Lines 161 – 164.

Regarding livestock abortion in Tanzania, as explained in the Introduction (lines 112-114), there is little data on abortion in livestock in Tanzania and elsewhere. Nonetheless, in the Discussion, we do describe the results with respect to other abortion studies carried out in

Ethiopia, Nigeria and India (lines 592-598). Moreover, as described in the Introduction (line 90-94), the implementation of syndromic or event-based surveillance in livestock is rare and to the authors’ knowledge has mostly been implemented in Europe, North America or Australasia with only a single pilot project identified in Africa.

494: Why will you use an estimate for abortion event that were not reported

As described above, this comment reflects a misunderstanding of what was being described. As written in line 494, an attempt was made to gauge the sensitivity of the surveillance platform by estimating the percentage of expected abortions that the investigated cases represented. That is, to estimate the percentage of abortions that the surveillance platform managed to detect, we divided the number of investigated abortions by the expected number of abortions (in each ward). The method for this estimation was described in lines 228-238.

511: Why was farming pattern excluded. Livestock rearing condition is equally critical for this type of investigation example an animal reared intensive system farming method will definitely experience different stress than livestock on nomadic free range system

We agree with the reviewer that livestock rearing system might be expected to impact both the aetiology and incidence of livestock abortion. However, because the number of wards was small and the distribution across system not equal, any association between investigated cases and livestock rearing system could not be assessed. We have made this clearer with additional text in the same paragraph of the Discussion.

529: Nothing was mentioned about educating the farmers or livestock owners to assist in some instances on possible sample collection during this abortion events and

sending these samples as quickly as possible to the central laboratory in suitable condition for investigation and result of the finding communicated back to the farmers

Because abortions can be caused by zoonotic pathogens, we did not involve livestock keepers in the collection of samples. Rather, sample collection was carried out by the research team and livestock field officers who had received appropriate training. In addition, results were reported back to the livestock keepers within 10 days of the investigation and, where pathogens were detected, more specific advice provided as to management strategies that could minimise further transmission to livestock and people. This is all described in the Methods (lines 181-199).

540: The livestock owner can be taught how to collect vaginal swab and send samples under suitable condition to the laboratory and the findings reported back to them.

Please see above response.

549: Please summerise.

Line 549-581 succinctly describes the attribution of cases to specific pathogens. The text given is required for comprehension and any further summarisation could impact understanding. Consequently, we have left the text as it is.

584: Please summerise.

Line 584-626 describes the patterns of livestock abortion in Tanzania. The text given is required to fully discuss the findings and any further reduction in text could impact understanding. Consequently, we have left the text as it is.